# Diversity of the Maize Root Endosphere and Rhizosphere Microbiomes Modulated by the Inoculation with *Pseudomonas fluorescens* UM270 in a Milpa System

**DOI:** 10.3390/plants13070954

**Published:** 2024-03-26

**Authors:** Blanca Rojas-Sánchez, Hugo Castelán-Sánchez, Esmeralda Y. Garfias-Zamora, Gustavo Santoyo

**Affiliations:** 1Genomic Diversity Lab, Institute of Chemical and Biological Research, Universidad Michoacana de San Nicolas de Hidalgo, Morelia 58030, Mexico; 1159358b@umich.mx (B.R.-S.); hcastelans@gmail.com (E.Y.G.-Z.); 2Department of Pathology and Laboratory Medicine, Western University, London, ON N6A 3K7, Canada; 1703153e@umich.mx

**Keywords:** bioinoculants, PGPR, milpa system, plant bacteriome, endophytes

## Abstract

Milpa is an agroecological production system based on the polyculture of plant species, with corn featuring as a central component. Traditionally, the milpa system does not require the application of chemicals, and so pest attacks and poor growth in poor soils can have adverse effects on its production. Therefore, the application of bioinoculants could be a strategy for improving crop growth and health; however, the effect of external inoculant agents on the endemic microbiota associated with corn has not been extensively studied. Here, the objective of this work was to fertilize a maize crop under a milpa agrosystem with the PGPR *Pseudomonas fluorescens* UM270, evaluating its impact on the diversity of the rhizosphere (rhizobiome) and root endophytic (root endobiome) microbiomes of maize plants. The endobiome of maize roots was evaluated by 16S rRNA and internal transcribed spacer region (ITS) sequencing, and the rhizobiome was assessed by metagenomic sequencing upon inoculation with the strain UM270. The results showed that UM270 inoculation of the rhizosphere of *P. fluorescens* UM270 did not increase alpha diversity in either the monoculture or milpa, but it did alter the endophytic microbiome of maize plant roots by stimulating the presence of bacterial operational taxonomic units (OTUs) of the genera *Burkholderia* and *Pseudomonas* (in a monoculture), whereas, in the milpa system, the PGPR stimulated greater endophytic diversity and the presence of genera such as *Burkholderia*, *Variovorax*, and N-fixing rhizobia genera, including *Rhizobium*, *Mesorhizobium*, and *Bradyrhizobium*. No clear association was found between fungal diversity and the presence of strain UM270, but beneficial fungi, such as *Rizophagus irregularis* and *Exophiala pisciphila*, were detected in the Milpa system. In addition, network analysis revealed unique interactions with species such as *Stenotrophomonas* sp., *Burkholderia xenovorans*, and *Sphingobium yanoikuyae*, which could potentially play beneficial roles in the plant. Finally, the UM270 strain does not seem to have a strong impact on the microbial diversity of the rhizosphere, but it does have a strong impact on some functions, such as trehalose synthesis, ammonium assimilation, and polyamine metabolism. The inoculation of UM270 biofertilizer in maize plants modifies the rhizo- and endophytic microbiomes with a high potential for stimulating plant growth and health in agroecological crop models.

## 1. Introduction

Milpa is a traditional open-field polyculture system that is still preserved as the main production system in various regions of Mexico and Latin America. It consists of the rotation of several plant species, with corn (*Zea mays* L.) featuring as the central crop, and it may include other plant crops such as Mexican husk tomatoes (*Physalis* spp.), common beans (*Phaseolus vulgaris* L.), pumpkins (*Cucurbita* spp.), and others [1]. The milpa system usually does not require the input of agrochemicals; therefore, its production depends on its own ecological resources (e.g., recycling of organic matter and biological control mechanisms). It can be affected by potential pathogens in addition to being grown in soils that can be nutritionally poor and very irregular in their orography, such as those that exist in the southeastern region of Mexico. The milpa, like other crop rotation systems [2], is a system that favors synergy between different species, as well as their short- and long-term rotation, stimulating better overall yields and generating resilience to external disturbances such as attack by pathogens and stressful abiotic conditions [3].

Likewise, by having several vegetable crops, milpa can generate greater species richness [2,3,4], and thisinvolves microorganisms in the soil and rhizosphere zones. Recently, Ariza-Mejía et al. [5] evaluated the rhizosphere diversity of two Physalis species (ixocarpa and philadelphica), maize grown in milpa, and bulk soil, finding a wide diversity of bacterial genera associated with *Physalis*, such as *Nocardioides*, *Streptomyces*, *Pseudonocardia*, and *Solirubrobacter*. On the other hand, the microbiome associated with corn plants has been widely analyzed under different environmental conditions (e.g., pH or soil type) [6,7], genotypes/varieties [8], and interaction zones, such as the rhizosphere [9,10], endosphere [11], and phyllosphere [12], among others. From these studies, it has been determined that the structure of the microbial communities of maize in the rhizosphere is highly dependent on the genotype of the plant, and its variation can also be modified by other factors, such as organic and inorganic fertilization. This was confirmed by Peiffer et al. [9], who evaluated the bacterial diversity of the rhizosphere of 27 inbred varieties of modern maize, which exhibit wide genetic diversity when grown under field conditions. Based on this work, it was noted that bacterial groups such as Proteobacteria, Bacteroidetes, Actinobacteria, and Acidobacteria were among the most abundant and poorly heritable.

Plant growth-promoting rhizobacteria (PGPR) play an important role in agricultural systems such as biofertilizers, biostimulants, and bioprotectants [13]. In the case of maize, this crop has been used as a study model because of its importance worldwide as one of the most cultivated grains in the world. Therefore, there are multiple studies where PGPRs have been inoculated into corn crops, observing increases in their growth and production, even under stressful conditions such as drought [14]. Likewise, studies have shown that certain PGPRs can also protect corn from attack by pathogens, trough mechanisms like antibiosis (e.g., production of diffusible and volatile organic compounds), competition for spaces, nutrient deprivation, and 1-amino cyclopropane-1-carboxylic acid desaminase activity, in addition to stimulating immune defense mechanisms.

Another interesting topic to analyze is the impact of PGPR inoculation on the assembly and diversity of microbial communities associated with corn. For example, Ferrarezi et al. [15] recently evaluated the inoculation of a bacterial corsortium made up of *Bacillus thuringiensis* RZ2MS9 and *Burkholderia ambifaria* RZ2MS16, observing that it did not significantly alter the microbiome associated with corn. Similarly, the authors compared the inoculation of the consortium with the *Azospirillum brasilense* Ab-V5 strain, which is widely commercialized and applied to corn crops to increase production [16]. The authors concluded that there are multiple inconsistencies when expanding studies from greenhouse and field conditions; therefore, it is recommended to expand similar studies under different environmental conditions.

Despite multiple studies on the corn microbiome, the impact of PGPR inoculation on the composition and structure of the microbiota associated with different interaction zones, such as the rhizosphere and endosphere, is still not well understood. Therefore, in this study, the impact of the inoculation of the beneficial bacterium *P. fluorescens* strain UM270 on the root endophytic microbiome, as well as on the rhizobiome of corn plants in an open and polyculture system (such as the cornfield) was characterized. 

## 2. Results

### 2.1. Endobiome Analysis of Maize Roots

When inoculated into plant cultures, PGPR can modify the endophytic microbiome and stimulate the growth and fitness of the host. Thus, we evaluated whether the diversity and structure of the endobiome were modulated by the bioinoculation of maize plants in a monoculture system (maize roots + UM270) and in polyculture (maize roots + UM270 + Milpa system), using uninoculated maize roots as a control (Figure 1). The analysis was performed in triplicate using the composite samples.

The results suggested that *P*. *fluorescens* UM270 inoculation changed the endophytic microbiome of maize roots (Figure 1; maize roots + UM270) compared to uninoculated plants (Figure 1; maize roots treatment). Interestingly, maize roots inoculated with strain UM270 showed unexpected and very different endobiome diversities. Uninoculated maize roots showed a high abundance of OTUs belonging to the genera *Prosthecobacter* and *Curvibacter*, whose presence decreased in inoculated treatments. On the other hand, the bacterial OTUs of the genera *Burkholderia* and *Pseudomonas* were stimulated in a monoculture (Figure 1A; maize roots + UM270), whereas, in the milpa system (Figure 1A; maize roots + UM270 + Milpa system), the abundance of plant-associated genera, such as *Burkholderia, Variovorax*, and N-fixing rhizobia genera, such as *Rhizobium, Mesorhizobium*, and *Bradyrhizobium*, increased. 

Figure 1B shows the fungal diversity found in the maize roots from each treatment, including those biofertilized with UM270, either in mono- or polyculture (Milpa system). As noted, no significant association was correlated with the presence of the UM270 strain; however, it was interesting to detect a high abundance of mycorrhizal fungi, such as Rizophagus irregularis or the plant growth-promoting fungus *Exophiala pisciphila*. 

The increase in the number of these OTUs was better observed in Figure 2 (panels A and B) for bacteria and fungi, respectively. Some OTUs, shown in gray color, unexpectedly increased in the milpa system (Treatment 3, maize roots + UM270 + Milpa system), which belong to *Burkholderia* and *Variovorax* genera. Other N-fixing bacteria were also increased in plants inoculated with the UM270 strain, but not at the same level detected in the genera *Burkholderia* and *Variovorax*. It was also noted that some OTUs, such as *Candidatus* Phytoplasma (a phytoplasma taxon associated with aster yellows disease), were also increased in one of the composed samples. However, no disease symptoms were detected in the maize plants.

Figure 3A,B also show significant differences at the genus level of the OTUs found in the diversity of the *endo*bacteriom*e* that was modulated by the interaction with the PGPR UM270. For example, in the top five genera modulated, *Prosthecobacter*, *Burkholderia*, *Pseudomonas*, *Rhizobium*, and *Variovorax* were found, whereas a decrease in the relative abundance of *Prosthecobacter* was observed in the maize roots + UM270 treatment of 2.6-fold, while in the *Maize* roots + UM270 + Milpa system, there was a decrease of 5.6-fold. In contrast, in the other genera, there was a increase in *Burkholderia* OTUs of 2.6-fold in the maize roots + UM270 treatment and a 1.5-fold change in the milpa system. The increase in *Pseudomonas* in the Maize roots + UM270 treatment of 7.5-fold increase is also surprising, while there was no change in the milpa. *Rhizobium* and *Variovorax* also showed an increase of approximately 2- and 5-fold, respectively, in the milpa treatment. Such differences in the top five genera demonstrated a significant difference relative to the control plants (uninoculated) according to the χ^2^ test (*p* < 0.05)*.*

In the case of the OTUs belonging to fungi, such evident results were not found when there was a correlation with the inoculation of the rhizobacterium *P. fluorescens* strain UM270. In contrast, the abundance of some possible species decreased with inoculation of the UM270 strain (such as in the case of the bacterial OTU *Prosthecobacter*). However, this hypothesis requires additional studies to detect certain antagonistic effects among the endophytic OTUs. 

### 2.2. Index Diversity Analysis

In this study, three of the main ecological indices were analyzed, including Shannon and Simpson indices, as shown in Figure 4, for bacterial and fungal endobiomes. The results showed that the inoculation of the UM270 bacterium in the monoculture and polyculture treatments of maize resulted in quite different alpha diversity, and also that the inoculation altered such measures. Although the alpha diversity in bacteria did not show an evident increase, this was not the case with the alpha diversity for fungi, where an increase in the evaluated indices was noted with respect to the control experiment, where there was no interaction with PGPR UM270.

Figure 5 shows the shared bacterial and fungal OTUs among the three treatment groups. The results showed that 41 bacterial and nine fungal OTUs were shared among the treatments; however, only two bacterial OTUs were found to be unique in maize roots without inoculation, and only one in roots inoculated with UM270 grown in a milpa system. No unique OTUs were found in the fungal endobiomes among treatments.

### 2.3. Endobiome Network Analysis

Network analysis was performed to evaluate possible species interactions among the three endobiomes (Figure 6). By identifying unique, common, and co-occurring species, we can better understand the potential ecological relationships among different species and their influence on the overall health and function of endobiomes. This information can be used to develop targeted interventions to promote a healthy endobiome and prevent imbalances in microbial communities [17].

Interestingly, despite the different maize treatments, multiple species were present in the endobiomes. This suggests that there are fundamental relationships between certain microbial species and maize plants that are unaffected by specific treatments.

### 2.4. Metagenomics of the Rhizosphere

Figure 7 shows the taxonomic profile of the rhizosphere metagenomes, including the maize roots (A1), maize roots + UM270 (A2), and maize roots + UM270 + Milpa systems (A3). In general, according to other analyses, no significant differences were found between the three treatments. The Proteobacteria group was the most abundant, followed by Firmicutes and Actibobacteria. However, when performing a heatmap analysis of functional activities detected in the microbial metagenomes based on SEED classifications, some differences were observed in the rhizospheres affected by the UM270 inoculation. For example, functions related to trehalose biosynthesis, ammonium assimilation, and polyamine metabolism are overrepresented in uninoculated maize roots. Figure 8 shows a heatmap of the different levels of analysis of metagenome functional annotations. 

## 3. Discussion

The results of this study show that the application of a biofertilizer based on the rhizobacterium *P. fluorescens* UM270 under the milpa model modulates the microbial diversity of root endophytes and rhizosphere microbiomes, and there was also a negative interaction between the inoculation of PGPR UM270 and the genus *Prosthecobacter*. *Prosthecobacter* has been associated with medicinal plants as an endophytic organism; however, the role of this bacterium, particularly the strains associated with plants, has been little explored [18]. In contrast, inoculation with beneficial microbial agents associated with plants can engage with other synergistic microbes [19]. Although the mechanism is not very clear, a recent study showed that pre-inoculation of pepper seedlings with the *Bacillus velezensis* strain NJAU-Z9 induced changes in the structure of the rhizosphere microbiome in a field experiment, stimulating communities of genera such as *Bradyrhizobium*, *Chitinophaga*, *Streptomyces*, *Lysobacter*, *Pseudomonas*, and *Rhizomicrobium* [20]. Recently, the endophytic bacteriome of *Medicago truncatula* was modified by the interaction of the biocompound N,N-dimethylhexadecylamine (DMHDA) produced by PGPRs such as *Arthrobacter* sp. UMCV2, and *Pseudomonas fluorescens* UM270. The results showed that bacterial groups such as β-proteobacteria and α-proteobacteria were more abundant in the root and shoot endophytic compartments, respectively [21]. Here, we observed that some genera, such as *Burkholderia* and *Variovorax*, and N-fixing rhizobia genera, such as *Rhizobium*, *Mesorhizobium* and *Bradyrhizobium*, were more abundant in maize cultures inoculated with rhizobacteria UM270. Therefore, it is also possible that these nitrogen-fixing bacteria were stimulated by nodulation factors released by bean plants, and, in turn, improved the acquisition of nitrogen, one of the elements that increased in the ear. It should be noted that it has been reported that intercropping between crops of faba beans (*Vicia faba* L.) and maize can result in overyielding and enhanced nodulation by faba beans [22]. Similarly, co-inoculation with *Rhizobium pisi* and *Pseudomonas monteilii* has been an effective biofertilization strategy for common bean production in Cuban soils [23]. Other studies have also shown synergism between rhizobia and PGPRs to increase the growth and production of maize and beans under different environmental conditions [24,25,26,27]. An increase in the abundance of nitrogen-fixing genera in the maize endophytic microbiome affected by inoculation with PGPR UM270 was not clearly detected in the rhizospheric microbiome, perhaps because of the capacity (and preference) of these rhizobia to colonize legume rhizospheres (such as *Phaseolus vulgaris*). Unfortunately, this is a limitation of our study; however, further studies are required to analyze other rhizospheres. 

The beneficial mycorrhizal fungus *Rizophagus irregularis* was potentially detected in the maize roots in the three treatments analyzed, including the milpa system. Some previous studies shows that *R. irregularis* can promote the growth of bean plants under greenhouse conditions, as well as under field conditions, having positive effects on maize, soybeans and wheat [28,29]. In 2022 [30], Chen and coauthors reported that *R. irregularis* is capable of modulating soil bacteriomes, in addition to modulating corn growth under salt stress conditions. In another study [31], a strain of *R. irregularis* was co-inoculated with a *Bacillus megaterium* strain, showing that the dual consortium improved maize tolerance to combined drought and elevated temperatures stresses by enhancing photosynthesis, root hydraulics, and regulating hormonal responses. Similarly, the endophytic fungus *Exophiala pisciphila*, particularly the H93 strain, has been an excellent promoter of plant growth in maize. One action of *E. pisciphila* is to improve plant nutrition by solubilizing phosphates [32]. Other species found as endophytes of maize were *Menispora tortuosa*, *Glyphium elatum* or *Phialocephala subalpina*, to mention a few, but they have been more associated with woody plants [33,34,35]; however, it would be interesting to explore its symbiotic functions with plants of agricultural interest.

Some of the bacterial species identified in this study were well known plant growth-promoting bacterial endophytes. It is present only in certain endobiomes, particularly in untreated maize with the PGPR UM270. These species include *Stenotrophomonas* sp., *Sphingobium yanoikuyae*, and *Burkholderia* spp. For example, *B. unamae* can use phenol and benzene as sole carbon sources; additionally, strains of *B. kururiensis* can metabolize trichloroethylene, 2,4,6-trichlorophenol and decompose phenol, benzene and toluene. Another strain of *B. tropica* degrades benzene, toluene, and xylene. Furthermore, the *B. xenovorans* strain LB400T is one of the most potent aerobic microorganisms and can degrade polychlorinated biphenyl (PCB). Some of these strains have been associated with crop plants, such as corn (in the case of *B. unamae* [36]). The unique occurrence of these bacteria suggests that they may play a role in corn plant growth, development, and health, particularly in the absence of external treatments. Interestingly, the same bacterial species were also detected in the rhizosphere metagenome. However, unlike the diversity found in the endospheric zone, no significant differences were observed in the rhizosphere. Therefore, diversity was very similar, with few differences. 

One of the common genera associated with maize plants is *Stenotrophomonas* sp., which has a wide range of metabolic capabilities and can survive under a variety of environmental conditions [37]. Some species of *Stenotrophomonas* have been found to be plant growth-promoting bacteria that can increase the growth and yield of crops such as maize. For example, some strains of *Stenotrophomonas* have been found to produce indole acetic acid, a plant hormone that stimulates the growth and development of maize roots.

*Burkholderia xenovorans* is another bacterial species found in the endobiomes of maize. This species degrades various environmental pollutants, including pesticides and herbicides. This suggests that *Burkholderia xenovorans* may play a role in detoxifying soil and protecting maize plants from the harmful effects of these chemicals [38].

*Sphingobium yanoikuyae* is a bacterial species that degrades polycyclic aromatic hydrocarbons (PAHs) and environmental pollutants that are toxic to plants [39]. This suggests that *Sphingobium yanoikuyae* may protect maize plants from the harmful effects of PAHs in soil. In the M2 condition (*maize + root UM270*), bacteria such as *Dyella marensis*, *Stenotrophomonas rhizophila*, and *Ralstonia* spp. co-occurred with M1 (*maize roots*), but not with M3 *(maize + UM270 + milpa system)*. The co-occurrence of *Dyella marensis*, *Stenotrophomonas rhizophila*, and *Ralstonia* spp. with M1, but not with M3, suggests that the addition of M3 may alter the microbial community structure in the maize endobiome and instead favor the development of other microbial species. *Dyella marensis* is a bacterial species that occurs in soil and is known for its ability to degrade a wide range of environmental pollutants. *Stenotrophomonas rhizophila* is another bacterial species known to promote plant growth, and it has been found in the endobiomes of several plant species, including corn. Some strains of *Stenotrophomonas rhizophila* produce plant hormones and enzymes that can stimulate root growth and plant development [37]. Plant growth-promoting properties have also been observed in some *Ralstonia* species, such as the production of plant hormones and enzymes that stimulate root growth and nutrient uptake. In the M3 system, *Pseudomonas putida*, *Pseudomonas thivervalensis*, and *Serratia fonticola* co-occurred only in M1 and not in M2. *Pseudomonas putida* is present in the endobiomes of several plant species, including maize, and may play a role in promoting plant growth and health [40]. *Pseudomonas thivervalensis* is a less well-studied bacterial species; however, some strains have been found to produce compounds that can inhibit the growth of plant pathogens [41]. *Serratia fonticola* is a bacterial species found in various environments, including soil and water [42].

Beta diversity detected in the endophytic microbiome of maize roots in monoculture and biofertilised with *P. fluorescens* UM270 showed the lowest biodiversity variation with respect to the other treatments. Although it can be argued that polyculture (or cropping practices) and fertilization with biological agents can stimulate greater endophytic diversity [43], the uninoculated maize monoculture also showed high variation. In general terms, it is important to highlight that, among the three treatments carried out in this work, the one associated with milpa is the most variable in terms of diversity and abundance, as all the triplicates vary from each other. In contrast, the most homogeneous triplicates were those of the inoculated “monoculture”. Similarly, it is important to point out that field experiments can generate wider variations than those performed under controlled conditions. However, the objective of this work was to get closer to the “reality” of field work, where abiotic conditions may not be so controlled, examining the inoculation of a bacterial agent, such as the UM270 strain, in such conditions in order to determine its performance.

As mentioned above, the taxonomic affiliations of the three rhizospheres analyzed in this study showed no significant differences. However, at the functional level, increases in trehalose biosynthesis, ammonium assimilation, and polyamine metabolism were observed. Trehalose (a-D-glucopyranosyl-1, 1-a-D-glucopyranoside) is a non-reducing disaccharide present in a wide variety of known organisms, some of which are known as anhydrobionts, including plants, fungi, and bacteria. Some plants can revive in the presence of water within a few hours of being completely dehydrated for months or years [44]. Trehalose-producing bacteria, such as rhizobia, can increase the biomass of maize and bean plants under drought conditions [45,46]. Similarly, ammonia assimilation is also related to nitrogen-fixing bacteria, such as *Rhizobium*, and its function seems to be relevant in this milpa system, where legume plants are co-cultivated with maize [47]. Enzymes such as Glutamine Synthetase (GS) and Glutamate Synthase (GOGAT) are important for the assimilation of ammonium; therefore, their search in rhizospheric soil in the cornfield would be relevant for determining their function in these environments. Polyamines play an important role in plant-bacteria communication, as well as in beneficial processes such as PGPR. In a recent review, Dunn and Becerra-Rivera [48] mentioned that polyamines are compounds that act as physiological effects and signal molecules in plant-bacteria interactions, so these functions can be found in rhizospheric environments modulated by the presence of *P. fluorescens* UM270 and could be an area that requires additional attention and research. Thus, the presence of PGPR plays an important role in its presence in rhizospheric environments, stimulating the synthesis of polyamines in other potentially beneficial microbes.

## 4. Materials and Methods

### 4.1. Experimental Site

The experiment was conducted in the town of Santa Clara del Cobre in the municipality of Salvador Escalante, Michoacán, Mexico. It is located at 19° 24′ 23″ North, 101° 38′ 24″ West, at an altitude of 2239 m. The prevailing climate is humid subtropical (Köppen climate classification: Cwa). 

Prior to the experiment, soil analysis was performed to determine its physicochemical characteristics. This analysis determined that the type of soil is clay and that it is composed of a percentage of 40% sand, 41.96% clay, and 18% silt.

### 4.2. Biological Material

Seeds of *Zea mays* L., *Phaseolus vulgaris* L., and *Cucurbita* spp. used in this experiment were obtained from the same municipality where the experiment was established and provided by local producers. The UM270 strain was used as a bioinoculant, and it was previously isolated and characterized [49].

### 4.3. Inoculum Preparation

Bacterial activation of *Pseudomonas fluorescens* strain UM270 was carried out by removing a hoe from the bacteria and placing it in a flask with 500 mL of Nutrient Broth (BD BIOXON, Franklin Lakes, NJ, USA), keeping it under constant agitation at 120 rpm at 28 °C for 24 h until an optical density (560–600 nm) of 1 was reached. The separation of the supernatant and the bacterial pellet was carried out to subsequently suspend it in solution with 0.1 mM magnesium sulfate (MgSO_4_), while a count of colony forming units (CFU) per milliliter was carried out during serial dilutions on Nutrient Agar media (BD BIOXON).

### 4.4. Seed Treatments

Seed preparation consisted of a superficial disinfection process involving washing with 70% ethanol, 5% sodium hypochlorite, and sterile distilled water [50]. The seeds used for the treatments in the presence of the bacterial strain were inoculated at a concentration of approximately 1 × 10^3^ CFU per seed. Control seeds were inoculated with MgSO_4_ solution only. The standard deviation of each inoculum was never greater than 10%. The average CFU per seed was extended in triplicate experiments (three seeds/replication) in which a seed with bacterial inoculum was placed by immersion in a nutritious liquid culture (5 mL). After vortexing, dilutions were made in nutrient agar at 28 °C for 48 h to quantify CFU/seed. Nine seeds were analyzed during serial dilution. 

### 4.5. Establishment of the Experiment under Field Conditions

The land preparation was carried out using the minimum essential procedures, which consist of clearing the land, followed by fallowing, then a pass with a harrow and furrowing, aiming to not overturn the surface layer of the soil, using animal traction. After this traditional task, maize planting was carried out on 11 May 2021, and the entire stage of cultivation ended in December of the same year. Native maize seeds known as “white maize” were used. This variety was selected for its nixtamalization and tortilla flavor characteristics. After two weeks, guide beans and pumpkins were planted. One month after planting the maize, a second inoculation with the *P. fluorescens* UM270 strain at a concentration of 1 × 10^8^ UFC was performed on the crops with the inoculated seeds, and, after another month, a third inoculation was performed at the same concentration. The *P. fluorescens* UM270 inoculations were applied in liquid form between 10 and 20 cm from the stem of each maize plant.

### 4.6. Experimental Design

The experimental design was completely randomized, featuring three treatments in which three crops were planted at different planting densities. According to the recommendations of the producers in the region, eight maize plants per m^2^ were planted, with 100 plants in each treatment. The composition of the polycultures was calculated as follows: planting a maize plant is equivalent to 0.75 bean plants and 0.25 pumpkin plants. The treatments evaluated were: (1) *Zea mays* L. (maize roots); (2) *Zea mays* L. + UM270 (maize roots + UM270); and (3) *Zea mays* L. + UM270 + *Phaseolus vulgaris* L. + *Cucurbita* spp. (maize roots + UM270 + Milpa system).

### 4.7. Endophytic DNA Extraction and Illumina Sequencing

Three samples composed of ten healthy maize plant roots (1 g of lateral root tissue from each plant) were pooled to isolate genomic DNA and sequence the endophytic microbiome, including bacteria and fungi. Briefly, soil particles were removed and root tissues were washed and superficially sterilized by immersion in 70% ethanol for 30 s, then in a 2.5% solution of commercial bleach for 5 min, followed by at least five times washing with sterile distilled water. To further confirm the sterilization process, an aliquot from the last rinse of sterile distilled water was cultured on plates with a nutrient agar medium and incubated at 28 °C for 72 h. No growth of bacterial or fungal colonies was observed in the plates after incubation. Then, plant root tissues were macerated using mortars in liquid nitrogen under sterile conditions, following the DNA extraction protocol published by Mahuku (2004) [51], and further purified using a DNA purification kit (PROMEGA). The quantity and quality of the DNA were confirmed by electrophoresis on agarose gels stained with GelRed and visualized under UV light using a NanoDrop 1000 spectrophotometer (Thermo Scientific, Rockford, IL, USA). Nine samples (three from each treatment) with good quantity and purity were sequenced using the Illumina MiSeq platform at the Mr. DNA company (Houston, TX, USA). DNA libraries were constructed by amplifying the V3-V4 hypervariable region (Primers: 515F GTGYCAGCMGCCGCGGTAA; 806R GGACTACNVGGGTWTCTAAT) of the 16S rRNA gene and ITS regions (Pimers: ITS1F CTTGGTCATTTAGAGGAAGTAA; ITS2R GCTGCGTTCTTCATCGATGC) using Mr. DNA. Subsequently, these amplicons were tagged and attached to PNA PCR Clamps to reduce plastid/mitochondrial DNA amplification [52].

### 4.8. Data Processing 

The taxonomic levels of phyla and genera were examined and are indicated for the 16S rRNA gene and ITS sequences obtained with paired-end reads. The sequences were aligned and processed using a Parallel-META 3.5 workflow [53]. Operational taxonomic unit (OTU) clustering was performed using the SILVA database integrated into Parallel- META 3.5 using a 97% homology criterion. There must be at least two sequences: the minimum zero abundance criterion is 10%, and the average abundance threshold is 0.1%. The maximum and minimum abundances were set to 0.1% and 0%, respectively [53].

### 4.9. Analysis Alpha and Beta Diversities 

Statistical analyses of sequence richness and diversity were performed using the Simpson and Shannon estimators, respectively, implemented in the Phyloseq package (v1.42.0) [54]. In addition, taxonomic composition was visualized using boxplots and heatmaps using ampvis2 (v2.5.5) [55]. Beta-diversity was determined using Vegan (v2.6-4) [56]. 

### 4.10. Endobiome Network Analysis 

Endobiome network analysis involves the construction and analysis of networks that represent relationships between different species within the endobiome. For this analysis, we used the igraph library, a network analysis library for R. The library provides a wide range of tools and functions for network construction, analysis, and visualization.

### 4.11. Metagenomic DNA Isolation and Analysis of Soil Rhizosphere

Metagenomic DNA was isolated as previously described [57]. Briefly, Metagenomic DNA was extracted from the rhizospheric soil samples (*n* = 5) using the Mo Bio PowerSoil^®^ DNA Isolation Kit and further purified with the Mo Bio PowerClean DNA Cleanup Kit. The DNA was then quantified, and its quality was assessed using a NanoDropTM 2000 c spectrophotometer (Thermo Fisher Scientific, Waltham, MA, USA). The samples were sent to the Genomic Services Center of the MR DNA (Shallowater, TX, USA). Metagenomic analyses were conducted in a similar manner as previously published, following the same quality controls, assembly, and taxonomic and functional annotations [57]. 

### 4.12. Sequence Accession Numbers 

The raw sequences are available at NCBI under BioProject accession number PRJNA901513 and Sequence Read Archive (SRA) accession numbers SRR22351342, SRR22351344, SRR22351348, SRR22351343, SRR22351346, SRR22351345, SRR22351347, and SRR22351341.

### 4.13. Statistical Analysis

The data obtained were analyzed by analysis of variance, and the variables that presented significant differences were analyzed by Tukey’s test (*p* ˂0.05) using the statistical package SAS (Statistical Analysis System) version 9.2.

## 5. Conclusions

Finally, the endobiome network allowed for the identification of different bacterial species present in the three maize treatment types, indicating the presence of fundamental relationships between certain microbial species and maize plants that were not affected by the specific treatments. In addition, some unique bacterial species have been identified in specific endobiomes (e.g., *Stenotrophomonas* spp. or *Burkholderia* spp.), some of which are also present in the rhizosphere, indicating their possible roles in the growth, development, and health of maize plants, especially in the absence of external treatments.

The addition of biofertilizers to maize plants grown under mild conditions, such as the *P. fluorescens* UM270 strain, modulates the rhizosphere and root endophytic microbiome. One of the potential mechanisms employed by the UM270 strain to stimulate plant growth may be the recruitment of other beneficial microorganisms through signaling molecules (e.g., polyamines). However, this hypothesis requires further investigation through the isolation and characterization of the synergistic activities of the inoculated strain UM270 and the associated microorganisms of maize plants.

## Figures and Tables

**Figure 1 plants-13-00954-f001:**
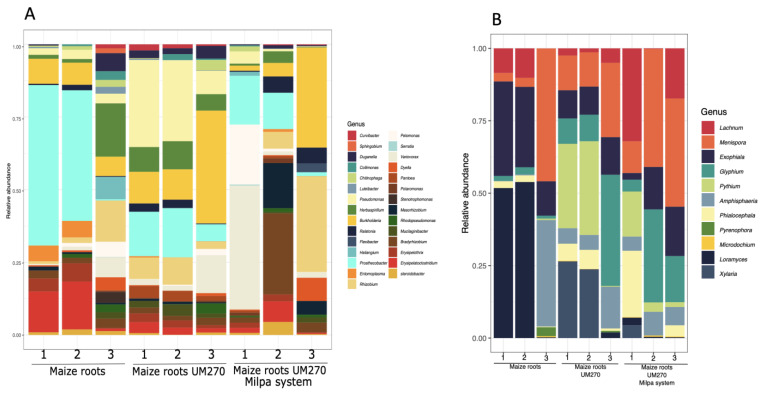
Relative abundances of bacterial (**A**) and fungal (**B**) taxa among the endophytic communities from maize plant roots cultivated in a milpa system.

**Figure 2 plants-13-00954-f002:**
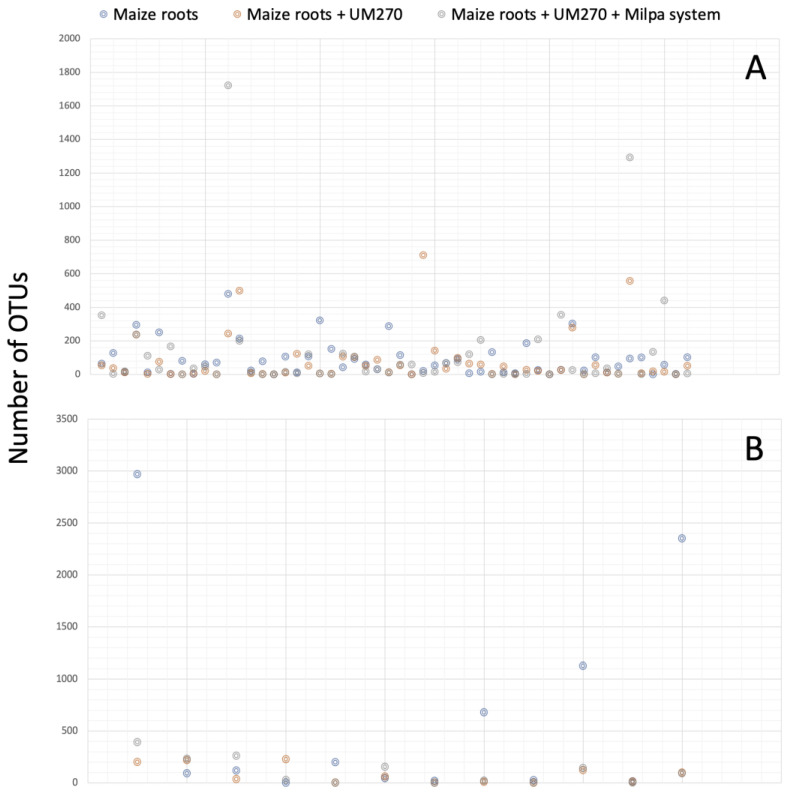
Number of bacterial (**A**) and fungal (**B**) OTUs detected in each of the three treatments.

**Figure 3 plants-13-00954-f003:**
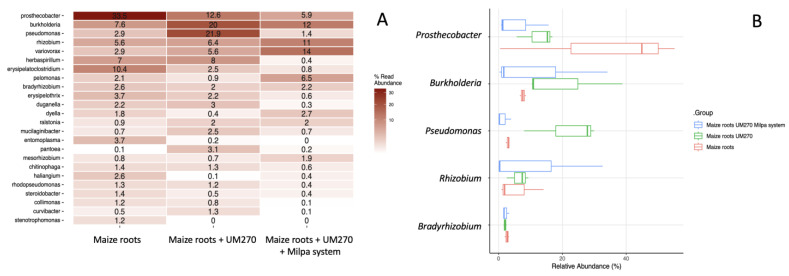
Heatmap of relative bacterial abundances of endophytic bacteriome detected in maize roots (Panel (**A**)), as well as the top five bacterial out treatments (Panel (**B**)).

**Figure 4 plants-13-00954-f004:**
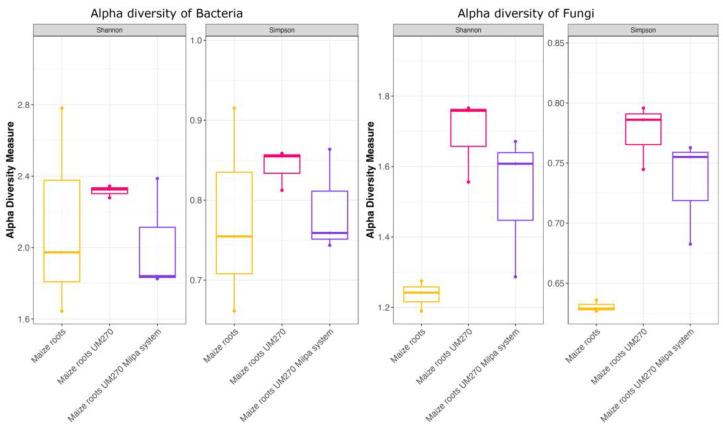
Alpha diversity indexes (measured by Shannon and Simpson diversity indexes) of bacterial and fungal endophytic communities from maize plant roots under three different experiments. Maize roots uninoculated (controls), maize roots inoculated with *P. fluorescens* UM270, and maize roots inoculated with *P. fluorescens* UM270 under a milpa system growth.

**Figure 5 plants-13-00954-f005:**
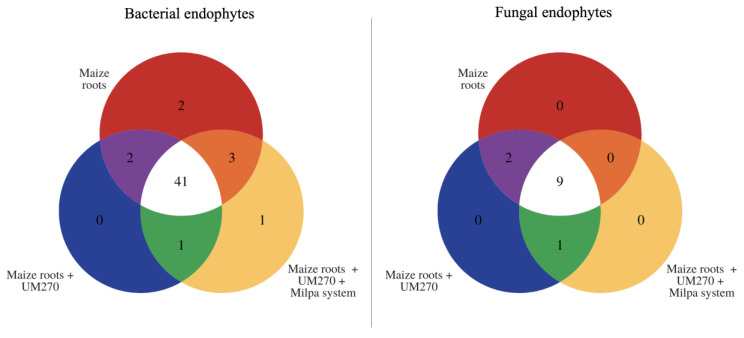
Shared OTUs among treatments. Regarding the unique OTUs found in the treatments where the UM270 strain was inoculated in a Milpa system, only one was found; On the other hand, there were no unique OTUs in corn roots inoculated with UM270. The maize root endobiome showed only two OTUs that were unique, while, for the diversity of fungal endophytes, no unique OTUs were found in each of the three treatments.

**Figure 6 plants-13-00954-f006:**
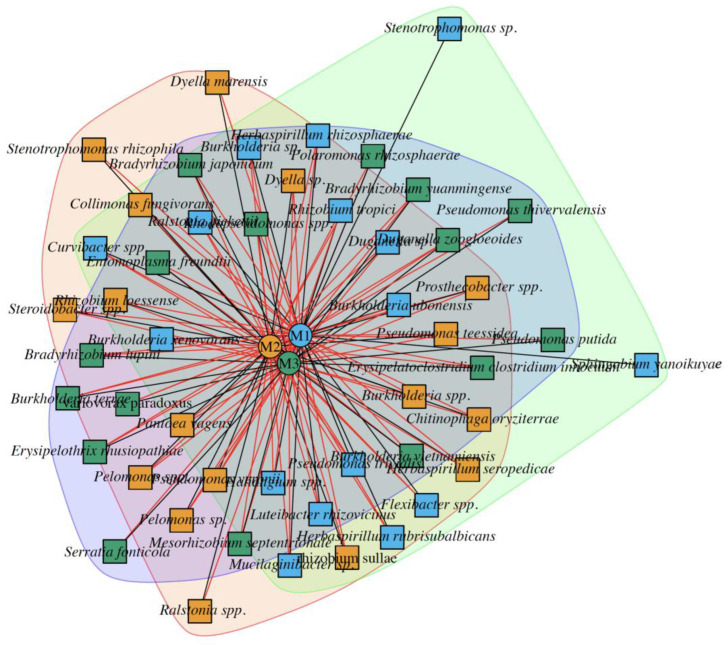
Network analysis of endophytic bacterial communities (endobiomes) from maize plant roots cultivated in a milpa system, inoculated or not with *P*. *fluorescens* UM270. The boxes represent individual endobiomes: M1 (maize roots), M2 (maize + root UM270), and M3 (maize + UM270 + milpa system). In the network, black lines indicate species that are unique and not present in the endobiomes. In contrast, the red lines indicate interactions or co-occurrences of species in the endobiomes.

**Figure 7 plants-13-00954-f007:**
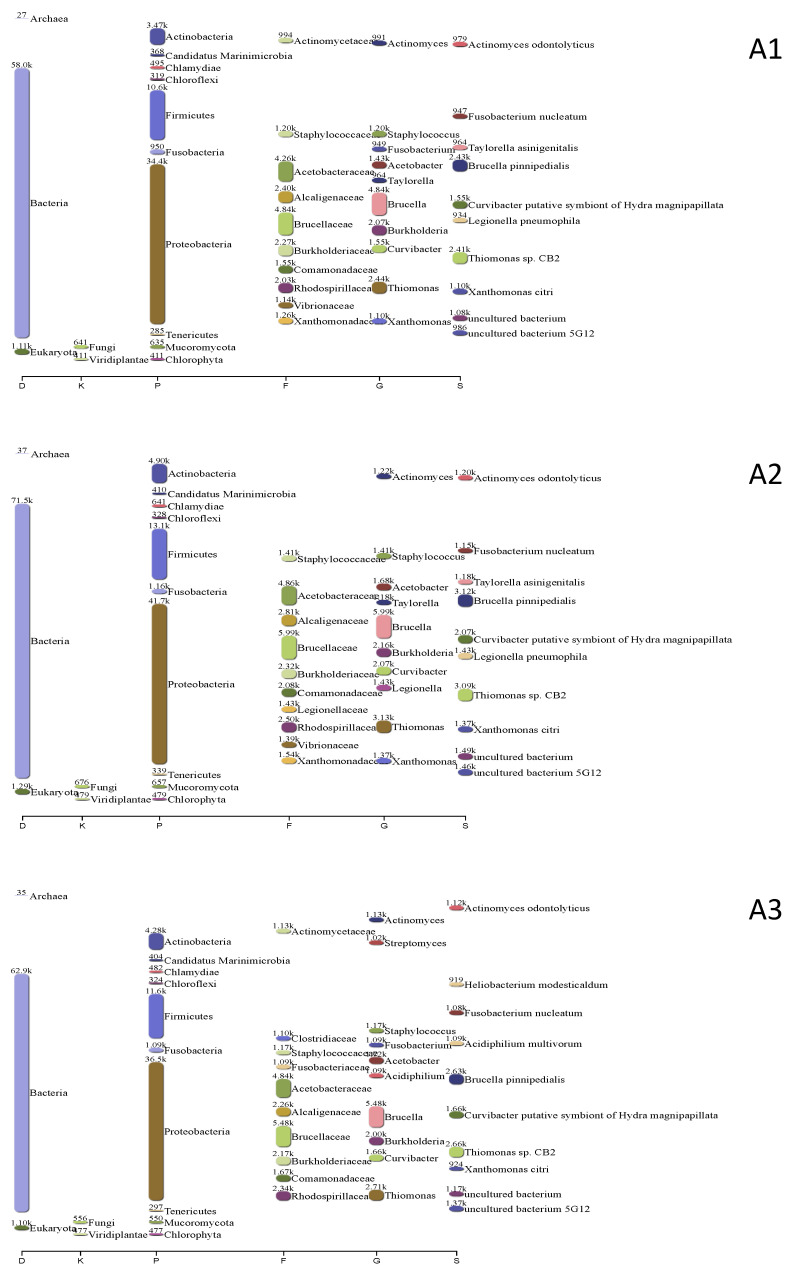
Taxonomic profiles of the rhizosphere metagenomes of maize (A1); maize inoculated with UM270 (A2), and maize inoculated with UM270 in a Milpa system (A3). The *x*-axis reports the taxonomic levels: D: domain; P: phylum; C: class; O: order; F: family; G: genus; S: species. Numbers correspond to the assigned contigs.

**Figure 8 plants-13-00954-f008:**
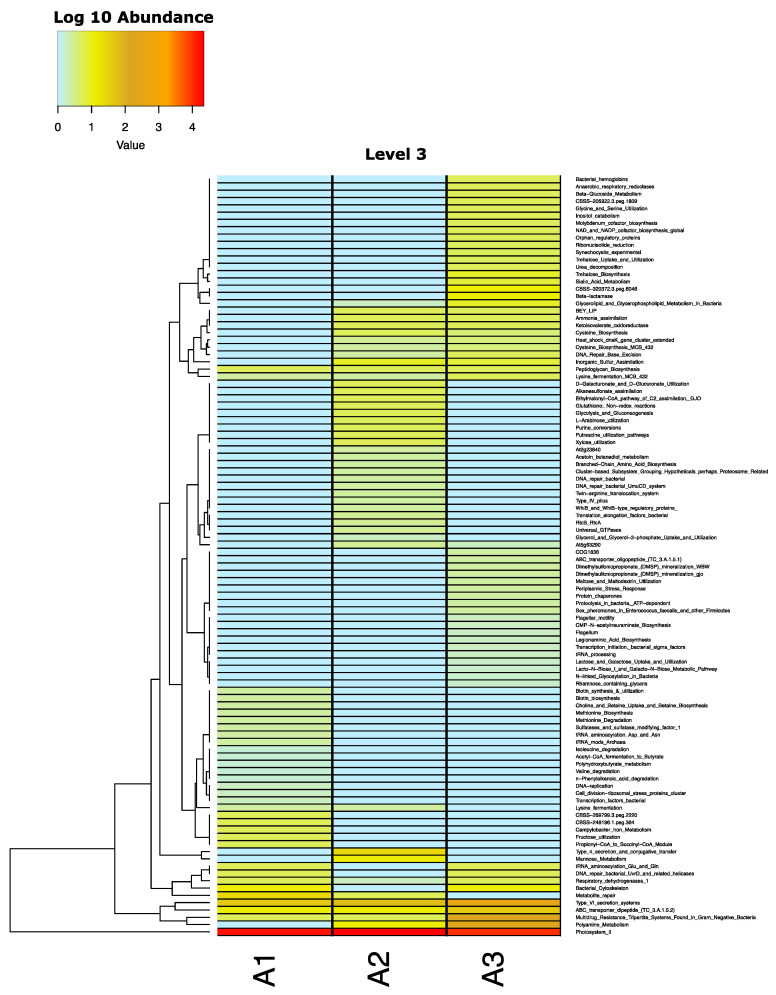
Heatmap of functional activities detected in the rhizosphere metagenomes of maize (A1); maize inoculated with UM270 (A2), and maize inoculated with UM270 in a Milpa system (A3).

## Data Availability

Sequencing data is freely available at the NCBI. See Materials and Methods.

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
