# Peer review of "Diversity of the Maize Root Endosphere and Rhizosphere Microbiomes Modulated by the Inoculation with Pseudomonas fluorescens UM270 in a Milpa System"

_plants, 2024, doi:10.3390/plants13070954_

Round 1

Reviewer 1 Report

Comments and Suggestions for Authors

Comments to the Authors

GENERAL COMMENTS

The manuscript titled: “Diversity of the maize root endosphere and rhizosphere microbiomes modulated by the inoculation with Pseudomonas fluorescens UM270 in a milpa system” - Manuscript ID: plants-2905527 is interesting and noteworthy.

Researchers aim to investigate the impact of inoculating the Plant Growth-Promoting Rhizobacteria (PGPR) Pseudomonas fluorescens UM270 on the rhizospheric and endophytic root microbiomes of maize crops in a milpa agroecological system. The study explores the potential benefits of using bioinoculants, focusing on their effects on microbial diversity and functions associated with maize growth and health. The results indicate alterations in the endophytic microbiome, stimulation of specific bacterial genera, and unique interactions in the rhizosphere. The researchers suggest that the UM270 strain, while not significantly affecting microbial diversity, plays a crucial role in functions like trehalose synthesis, ammonium assimilation, and polyamine metabolism. Overall, the study aims to understand how the application of UM270 biofertilizer modifies microbiomes, contributing to improved plant growth and health in agroecological crop models.

The research results obtained are valuable, however, in my opinion, the manuscript needs some, minor changes, and improvements. First of all, the introduction needs a redrafting. Materials and methods are described correctly. The greatest value of the manuscript is statistical analyses, and graphic solutions, which in particular increase its value. The discussion should also be improved, and enriched with appropriate content.

SPECIFIC COMMENTS

Abstract

"In my opinion, the abstract lacks a clearly outlined research objective

Introduction

The introduction is well-crafted in terms of content. However, it lacks an elaborated research objective and stated research hypotheses in its final part.

Discussion

Page 10, line 239

In my opinion, you can highlight the practical implications of the findings for agriculture, such as how PGPR inoculation affects nitrogen availability for plants, which can impact maize yield.

Page 10, line 247

When discussing Rizophagus irregularis, you could elaborate on its specific mechanisms of interaction with maize. Highlighting its role in nutrient uptake or stress tolerance could add depth to the understanding of its impact on plant growth. Describe it in just a few sentences.

Page 10, line 254

Provide a brief explanation of how Exophiala pisciphila, especially strain H93, enhances plant growth in maize, particularly under stress conditions induced by heavy metals. Mentioning its action in phosphorus solubilization could underscore its importance in plant nutrition.

Page 10, line 272

Burkholderia xenovorans in soil detoxification: In the context of Burkholderia xenovorans, it is worth adding information about specific chemicals that this species degrades and the potential consequences for soil and maize plants. Describe it in just a few sentences.

Author Response

Dear Reviewer, 

We would like to thank you for your critical comments and suggestions, that for sure, have improved our manuscript.

Sincerely,

Prof. Dr. Gustavo Santoyo

We are answering as follows:

GENERAL COMMENTS

The manuscript titled: “Diversity of the maize root endosphere and rhizosphere microbiomes modulated by the inoculation with Pseudomonas fluorescens UM270 in a milpa system” - Manuscript ID: plants-2905527 is interesting and noteworthy.

Researchers aim to investigate the impact of inoculating the Plant Growth-Promoting Rhizobacteria (PGPR) Pseudomonas fluorescens UM270 on the rhizospheric and endophytic root microbiomes of maize crops in a milpa agroecological system. The study explores the potential benefits of using bioinoculants, focusing on their effects on microbial diversity and functions associated with maize growth and health. The results indicate alterations in the endophytic microbiome, stimulation of specific bacterial genera, and unique interactions in the rhizosphere. The researchers suggest that the UM270 strain, while not significantly affecting microbial diversity, plays a crucial role in functions like trehalose synthesis, ammonium assimilation, and polyamine metabolism. Overall, the study aims to understand how the application of UM270 biofertilizer modifies microbiomes, contributing to improved plant growth and health in agroecological crop models.

The research results obtained are valuable, however, in my opinion, the manuscript needs some, minor changes, and improvements. First of all, the introduction needs a redrafting. Materials and methods are described correctly. The greatest value of the manuscript is statistical analyses, and graphic solutions, which in particular increase its value. The discussion should also be improved, and enriched with appropriate content.

RESPONSE: Thank you very much for your comments and valuable revision.

SPECIFIC COMMENTS

Abstract

"In my opinion, the abstract lacks a clearly outlined research objective

 RESPONSE: Thank you for your suggestion. Modified as suggested.

Introduction

The introduction is well-crafted in terms of content. However, it lacks an elaborated research objective and stated research hypotheses in its final part.

 RESPONSE: Thank you for your suggestion. Modified as suggested.

Discussion

Page 10, line 239

In my opinion, you can highlight the practical implications of the findings for agriculture, such as how PGPR inoculation affects nitrogen availability for plants, which can impact maize yield.

RESPONSE: We appreciate your comment. We believe this is discussed in the lines P10, L240-245.

Page 10, line 247

When discussing Rizophagus irregularis, you could elaborate on its specific mechanisms of interaction with maize. Highlighting its role in nutrient uptake or stress tolerance could add depth to the understanding of its impact on plant growth. Describe it in just a few sentences.

RESPONSE: Thank you for your suggestion. We added two more references and the following paragraph: ¨ In 2022 [30], Chen and coauthors reported that R. irregularis is capable of modulating soil bacteriomes, in addition to modulating corn growth under salt stress conditions. In another study [31], a strain of R. irregularis was co-inoculated with a Bacillus megaterium strain, showing that the dual consortium improved maize tolerance to combined drought and elevated temperatures stresses by enhancing photosynthesis, root hydraulics, and regulating hormonal responses.¨

References added:

  1. Chen, Q.; Deng, X.; Elzenga, J.T.M.; van Elsas, J.D. Effect of soil bacteriomes on mycorrhizal colonization by Rhizophagus irregularis—interactive effects on maize (Zea mays L.) growth under salt stress. Biol. Fertil. Soils 2022, 58, 515–525, doi:10.1007/s00374-022-01636-x.
  2. Romero-Munar, A.; Aroca, R.; Zamarreño, A.M.; García-Mina, J.M.; Perez-Hernández, N.; Ruiz-Lozano, J.M. Dual Inoculation with Rhizophagus irregularis and Bacillus megaterium Improves Maize Tolerance to Combined Drought and High Temperature Stress by Enhancing Root Hydraulics, Photosynthesis and Hormonal Responses. Int. J. Mol. Sci. 2023, 24, doi:10.3390/ijms24065193.

Page 10, line 254

Provide a brief explanation of how Exophiala pisciphila, especially strain H93, enhances plant growth in maize, particularly under stress conditions induced by heavy metals. Mentioning its action in phosphorus solubilization could underscore its importance in plant nutrition.

 RESPONSE: Thank you for your observation, our mistake. We checked again the manuscript, and this is not stated in such report. The sentence ¨ particularly under stress conditions induced by heavy metals¨, was deleted.

Page 10, line 272

Burkholderia xenovorans in soil detoxification: In the context of Burkholderia xenovorans, it is worth adding information about specific chemicals that this species degrades and the potential consequences for soil and maize plants. Describe it in just a few sentences.

RESPONSE: Great comment. We added the following sentences: ¨ Some of the bacterial species identified in this study were. It is present only in certain endobiomes, particularly in untreated maize with the PGPR UM270. These species include Stenotrophomonas sp.,  Sphingobium yanoikuyae, and Burkholderia spp. For example, B. unamae can use phenol and benzene as sole carbon sources; Also, strains of B. kururiensis can metabolize trichloroethylene, 2,4,6-trichlorophenol and decompose phenol, benzene and toluene. Another strain of B. tropica degrades benzene, toluene, and xylene. Furthermore, B. xenovorans strain LB400T is one of the most potent aerobic microorganisms that degrade polychlorinated biphenyl (PCB). Some of these strains have been associated with crop plants, such as corn in the case of B. unamae

The following reference was added: Estrada-De Los Santos, P., Rojas-Rojas, F. U., Tapia-García, E. Y., Vásquez-Murrieta, M. S., & Hirsch, A. M. (2016). To split or not to split: an opinion on dividing the genus Burkholderia. Annals of Microbiology, 66, 1303-1314.

Reviewer 2 Report

Comments and Suggestions for Authors

Generally, the study is interesting with potential to have an impact on several areas of crop development. However, there are a lot of missing details in the manuscript that need to be addressed. 

The introduction provides a good background to the theoretical underpinning of the study. However, there is a lot of chatty and unprofessional language used, as well as used throughout the article. For example, line 78. This needs to be addressed.

The results section has too much discussion in it and does not follow the correct scientifically accepted formatting.

Figure 1 a and b need to be larger.

Figure legends, like figure 5, need more information in them. The treatments needs to be stated in the legend as well as the figure itself.

There is very little statistical analysis.

It's not possible to identify R. irregularis from ITS sequencing alone. It is highly unlikely you identified this fungus. Illumina sequencing cannot reliably identify to species level. How are you making species assumptions? Other primer sequences are needed to achieve this. What are the confidence intervals? More details in the methods section would help to resolve this.

Line 314 - You mention no significant differences, but there is little stats to support this.

The journal's article formatting needs to be followed correctly. The methods section is missing the correct section heading and naming formats.

All manufacturer details must be provided in the methods section.

Why was 1x10^3 cfu used? This is not justified anywhere. Typically, 1x10^4 cfu's are used for seed treatment. This difference needs to be addressed in the article.

What are the stages of seed surface sterilisation? You've said what the solutions are but not provided the required details of the process. Your methods are not repeatable in their present form. At the very least a reference is needed.

Line 396 - What are the traditional techniques? This is important information to have stated.

What are the forward and reverse 16s and ITS primers used? These details need to be stated.

The methods section need a lot of work with the required details adding. Currently, none of these methods can be reproduced due to lack of detail. This section is also unclear and unstructured.

Comments on the Quality of English Language

Grammar and general English needs to be proof read

Author Response

Dear Reviewer, 

We would like to thank you for your critical comments and suggestions, that for sure, have improved our manuscript.

Sincerely,

Prof. Dr. Gustavo Santoyo

We are answering as follows:

Generally, the study is interesting with potential to have an impact on several areas of crop development. However, there are a lot of missing details in the manuscript that need to be addressed. 

RESPONSE: Thank you for your time to review our work and critical comments.

The introduction provides a good background to the theoretical underpinning of the study. However, there is a lot of chatty and unprofessional language used, as well as used throughout the article. For example, line 78. This needs to be addressed.

RESPONSE: Thank you very much for your comment. The sentence was modified and the text was further checked. ¨… certain PGPRs can also protect corn from attack by pathogens trough mechanisms like antibiosis (e.g. production of diffusible and volatile organic compounds), competition for spaces, nutrient deprivation, and 1-amino cyclopropane-1-carboxylic acid desaminase activity, to mention but a few), in addition to stimulating immune defense mechanisms.¨…

The results section has too much discussion in it and does not follow the correct scientifically accepted formatting.

RESPONSE: Thank you, we checked again our results but we think it is necessary to better describe the results.

Figure 1 a and b need to be larger.

RESPONSE: Thank you, you are right! Modified as suggested.

Figure legends, like figure 5, need more information in them. The treatments needs to be stated in the legend as well as the figure itself.

RESPONSE: Thank you for your suggestion, the legend of Figure 5 was better described as follows: ¨ Figure 5. Shared OTUs among treatments. Regarding the unique OTUs found in the treatments where the UM270 strain was inoculated in a Milpa system, only one was found; On the other hand, there were no unique OTUs in corn roots inoculated with UM270. The maize root endobi-ome showed only two OTUs that were unique, while for the diversity of fungal endophytes, no unique OTUs were found in each of the three treatments.¨.

There is very little statistical analysis.

RESPONSE: We believed the statistics is well described in section: ¨ Statistical analysis: The data obtained were analyzed by analysis of variance, and the variables that presented significant differences were analyzed by Tukey's test (p Ë‚0.05), using the sta-tistical package SAS (Statistical Analysis System) version 9.2.¨. In addition, we also evaluated diversity indexes, such as Alpha diversity indexes (measured by Shannon and Simpson diversity indexes) of bacterial and fungal endophytic communities from maize plant roots under three different experiments.

It's not possible to identify R. irregularis from ITS sequencing alone. It is highly unlikely you identified this fungus. Illumina sequencing cannot reliably identify to species level. How are you making species assumptions? Other primer sequences are needed to achieve this. What are the confidence intervals? More details in the methods section would help to resolve this.

RESPONSE: You are right; however, we did not state this. We handle read sequences as bacterial and fungal OTUs, which could give us an inference of the potential bacterial and fungal species that reside or are associated to maize roots and rhizosphere.  

Line 314 - You mention no significant differences, but there is little stats to support this.

RESPONSE: Our analysis showed no correlation in some of our analysis; however, we consider that these negative results are not worthy to show in the text. We hope you understand.

The journal's article formatting needs to be followed correctly. The methods section is missing the correct section heading and naming formats.

RESPONSE: Thank you! Modified as suggested.

All manufacturer details must be provided in the methods section.

RESPONSE: Thank you! Modified as suggested.

Why was 1x10^3 cfu used? This is not justified anywhere. Typically, 1x10^4 cfu's are used for seed treatment. This difference needs to be addressed in the article.

RESPONSE: This is the number of CFUs recover after seed sowing in the bacterial inoculum. The inoculum contains more cells than that. Of course, this concentration could vary, that is why we used the word ¨approximately¨, as follows: ¨ … concentration of approximately 1X103 CFU per seed¨.

What are the stages of seed surface sterilisation? You've said what the solutions are but not provided the required details of the process. Your methods are not repeatable in their present form. At the very least a reference is needed.

RESPONSE: Thank you for your observation, we added a reference. Ortiz, M..; Hernández, J..; Valenzuela, B.; De Los Santo, S.; Del Carmen Rocha, M.; Santoyo, G. Diversity of cultivable endophytic bacteria associated with blueberry plants (Vaccinium corymbosum L.) cv. Biloxi with plant growth-promoting traits. Chil. J. Agric. Anim. Sci. 2018, 34, doi:10.4067/S0719-38902018005000403.

Line 396 - What are the traditional techniques? This is important information to have stated.

RESPONSE: Thank you! Modified as suggested.

What are the forward and reverse 16s and ITS primers used? These details need to be stated.

RESPONSE: Thank you! Modified as suggested. PRIMER SEQUENCES WERE ADDED.

The methods section need a lot of work with the required details adding. Currently, none of these methods can be reproduced due to lack of detail. This section is also unclear and unstructured.

RESPONSE: Thank you for your comment, however, we believe that these M&M are well-described. In addition, they are very common and can be reproduced by any Lab of Molecular Biology. A huge amount of similar papers also described these methodologies.  

Grammar and general English needs to be proof read

RESPONSE: Thank you for your comment. Our manuscript was evaluated by the professional editing services of Editage. If you further request a certificate, we can ask for it.   

Round 2

Reviewer 2 Report

Comments and Suggestions for Authors

Thank you for the responses and improvements made. A couple of further things to add:

It would be beneficial to add text to that of your previous response detailing the identification of R. irregularis from ITS sequencing. Your current results and text suggests that ITS can identify this fungus to species level. As we both know, it is not possible to do this. A line needs to be added to acknowledge this.

Out of curiosity, did you stain any root samples to check for mycorrhizal colonisation?

Line 391 should have species names

Author Response

Dear Reviewer,

attached you will find a revised copy of our work. Thank you again for your time and constructive comments.

We are answering as follows:

Thank you for the responses and improvements made. A couple of further things to add:

It would be beneficial to add text to that of your previous response detailing the identification of R. irregularis from ITS sequencing. Your current results and text suggests that ITS can identify this fungus to species level. As we both know, it is not possible to do this. A line needs to be added to acknowledge this.

RESPONSE: Thank you, we added the word ¨potentially¨to the sentence, making it clearer that it is not possible to be sure the species was identified by ITS sequencing.

Out of curiosity, did you stain any root samples to check for mycorrhizal colonisation?

RESPONSE: No, we did not. But certainly, your idea will be taken into account in future experiments!

Line 391 should have species names

RESPONSE: Modified as suggested.

Kind regards,

Gustavo Santoyo
